# tRNA Modifications: A Tale of Two Viruses—SARS-CoV-2 and ZIKV

**DOI:** 10.3390/ijms26157479

**Published:** 2025-08-02

**Authors:** Patrick Eldin, Laurence Briant

**Affiliations:** Institut de Recherche en Infectiologie de Montpellier (IRIM), University of Montpellier, CNRS UMR 9004, 1919 Route de Mende, 34293 Montpellier, France; laurence.briant@irim.cnrs.fr

**Keywords:** SARS-CoV-2, Zika, tRNA modifications, translation

## Abstract

tRNA modifications are crucial for efficient protein synthesis, impacting codon recognition, tRNA stability, and translation rates. RNA viruses hijack the host’s translational machinery, including the pool of modified tRNA, to translate their own genomes. However, the mismatch between viral and host codon usage can lead to a limited availability of specific tRNA leading to ribosome stalling, posing a significant challenge for efficient protein translation. While some viruses address this challenge through codon optimization, we show here that SARS-CoV-2 (*Coronavirus*) and the Zika virus (ZIKV; *Flavivirus*) adopt a different approach, manipulating the host tRNA epitranscriptome. Analysis of codon bias indices confirmed a substantial divergence between viral and host codon usage, revealing a strong preference in viral genes for codons decoded by tRNAs requiring U34 wobble modification. Monitoring tRNA modification dynamics in infected cells showed that both SARS-CoV2 and ZIKV enhance U34 tRNA modifications during infection. Strikingly, impairing U34 tRNAs profoundly impacted viral replication, underscoring the strict reliance of SARS-CoV-2 and ZIKV on manipulating the host tRNA epitranscriptome to support the efficient translation of their genome.

## 1. Introduction

Transfer RNA (tRNA) modifications are essential for protein synthesis and overall cellular function [1]. These modifications, which include methylation, thiolation, and other more complex forms [2], occur at various positions within the tRNA molecule, especially in the anticodon loop [3,4,5]. By influencing codon recognition, tRNA stability, and ribosome interactions, these modifications impact the efficiency and accuracy of translation [6]. They modulate the tRNA affinity for specific codons, stabilize tRNA structure, and regulate translation rates. The diverse combination of possible modifications allows for the precise tuning of tRNA function across different cellular contexts. Furthermore, tRNA modification levels are dynamically regulated in response to environmental changes and cellular stress, with some modifications even being actively reversed [7,8]. This dynamic regulation of tRNA modification contributes to cellular fitness and protein homeostasis under diverse conditions [3,9] and also plays a role in viral infections [10].

Among these modifications, the mcm^5^s^2^U (5-methoxycarbonylmethyl-2-thiouridine) modification at the U34 position (wobble base) of tRNA is critically important [11,12]. Eukaryotic cells use a single tRNA species (tRNA^Lys^UUU, tRNA^Gln^UUG, and tRNA^Glu^UUC) with a U at position 34 to decode both AA- and AG-ending codons (AAA/G for Lys, CAA/G for Gln, and GAA/G for Glu). However, efficient translation of the AA-ending codons depends significantly on the presence of the mcm^5^s^2^U modification at U34. This dependence on U34 modifications for proper decoding extends beyond Lys, Gln, and Glu to other amino acids with A-ending codons [13], including Arg (AGA/G), Ser (UCA/G), and Pro (CCA/G), whose respective tRNAs also require U34 modifications (mcm^5^U for tRNA^Arg^UCU; ncm^5^U for tRNA^Ser^UGA; and tRNA^Pro^UGG).

The redundancy of the genetic code, where multiple synonymous codons can specify a single amino acid, results in codon usage bias (CUB), which varies between organisms [14]. tRNA modifications, particularly those within the anticodon stem-loop region, are crucial for decoding certain synonymous codons and, in concert with a gene’s specific CUB, influence both the efficiency and accuracy of protein translation [15].

Viruses must use host cell machinery for protein synthesis, but differences between viral and host codon usage bias (CUB) can lead to inefficient translation due to limited tRNA availability. Some viruses have evolved to match their host’s CUB patterns, optimizing their protein production efficiency. Studies have demonstrated that viruses with narrow host ranges exhibit a higher degree of codon usage matching with their hosts compared to broad-spectrum viruses [16]. This adaptation process has been observed in canine parvovirus type 2 (CPV-2), which progressively aligned its CUB with that of its new canine host after jumping from felines [17]. Similar observations have more recently been made in the case of the Porcine epidemic diarrhea virus (PEDV) [18] and Avian Influenza A viruses (AIVs) [19] that both adapted their CUB to better fit to their respective host codon preferences. Coincidentally, where viral and host codon usage are inherently similar, as exemplified by poliovirus (PV) [20] and foot-and-mouth disease virus (FMDV) [21], competition for tRNAs for protein synthesis can attenuate viral translation [22]. Nevertheless, the vast majority of RNA viruses have not followed this evolutionary path. Instead, they have developed an alternative strategy, the focus of this report, based on the manipulation of tRNA modifications to enhance viral protein synthesis. We illustrate this strategy using two unrelated RNA viruses that recently emerged and triggered global threats: the Coronavirus SARS-CoV-2 and the Flavivirus Zika (ZIKV).

SARS-CoV-2, which caused the COVID-19 pandemic [23,24], has tragically affected global health, infecting over 600 million people and resulting in more than 6 million deaths (https://covid19.who.int, accessed on 10 January 2025). This betacoronavirus, with a positive-sense, single-stranded RNA genome of ~30 kb, is closely related to other human-infecting coronaviruses, including SARS-CoV-1 and MERS-CoV. Along with other betacoronaviruses (like HCoV-HKU1 and HCoV-OC43) and alphacoronaviruses (like HCoV-229E and HCoV-NL63), it belongs to the *Nidovirales* order. Given their shared classification, SARS-CoV-1 and MERS-CoV exhibit similar structural features to SARS-CoV-2 already detailed [25,26], and were thus included in downstream codon analyses.

ZIKV, a mosquito-borne *flavivirus*, causes serious neurological disorders in humans, including microcephaly in newborns. Discovered in Uganda in 1947 and initially causing sporadic cases, ZIKV later caused major outbreaks in the Pacific and South America, facilitated by the widespread *Aedes* mosquito vectors. The 2015 outbreak in Brazil rapidly spread throughout Latin America and the Caribbean [27,28,29]. ZIKV has a ~10.7 kb positive-sense single-stranded RNA genome encoding a polyprotein that is processed into structural and non-structural proteins. Our codon analysis will also include two related *Flaviviruses*: West Nile virus (WNV), spread by *Culex* mosquitoes and capable of infecting birds, humans, and horses worldwide, and Usutu virus (USUV), a more recent emergence in Africa and Europe, also primarily transmitted by *Culex* mosquitoes. While USUV predominantly affects birds, human infections are rare.

We first investigated the relationship between viral and host codon usage by comparing the viral Codon Adaptation Index (CAI) to that of human genes across a wide range of expression levels and further analyzing Relative Synonymous Codon Usage (RSCU). This approach revealed how much each virus’s codon bias differs from that of highly expressed human genes. Additionally, it uncovered the preference of both viruses for codons interpreted by cognate tRNAs carrying U34 modifications, referred to as U34-sensitive codons [25]. Concomitantly, we experimentally showed that SARS-CoV-2 and ZIKV infections induce a substantial increase in U34 tRNA modifications (ncm^5^U, mcm^5^U, and mcm^5^s^2^U), consistent with the high prevalence of U34-sensitive codons in the viral genomes. Finally, we further validated the critical dependence of both viruses on enhanced U34 tRNA modification through the observation of a profoundly decreased translation and attenuated replication upon the impairment of normal U34-modified tRNA generation in target cells.

This research represents a significant advancement in our understanding of viral strategies for survival and replication, potentially opening new avenues for therapeutic intervention in viral diseases.

## 2. Results

### 2.1. Analysis of SARS-CoV-2 and ZIKV Codon Bias

The Codon Adaptation Index (CAI) measures the extent to which a gene’s codon usage aligns with that of highly expressed genes within a given organism [30]. It is particularly useful for evaluating codon usage compatibility between a virus and its host, with values ranging from zero (indicating no adaptation) to one (signifying perfect adaptation to the host). We first compared the CAI values of coronaviruses and flaviviruses to those of human genes expressed at varying protein levels. As shown in Figure 1A, the CAI calculated for the entire viral genome suggests that neither SARS-CoV2 nor ZIKV are fully adapted to the human host. SARS-CoV-2, with a CAI of 0.701, aligns with human genes that are poorly expressed in human cells, while ZIKV, with a CAI of 0.756, shows a slightly better compatibility with the human environment. However, neither of these viruses achieve CAI values comparable to those of the most highly expressed human genes, such as beta-tubulin, GAPDH, or beta-myosin. Figure 1B shows the CAI values calculated for individual viral gene segments, revealing their deviation from the overall viral CAI. Notably, among the viral genes, E in SARS-CoV-2 and NS2B in ZIKV exhibit the lowest host adaptation, while N in SARS-CoV-2 and NS1 in ZIKV are adapted to humans. In addition, SARS-CoV-2 genes, as a whole, display lower Nc (total number of codons used) values than ZIKV genes, indicating a more restricted codon usage and, thus, a stronger codon usage bias.

The use of Relative Synonymous Codon Usage (RSCU) provides a more detailed view of codon usage bias compared to CAI [31], revealing the specific codons that each virus preferentially uses in comparison to its host [32]. Analysis of over-represented (RSCU > 1.6) and under-represented (RSCU < 0.6,) codons showed similar codon usage patterns within the Coronavirus and the Flavivirus families (Figure 1C). However, these patterns differed significantly from those of their respective hosts (humans for coronaviruses, and both humans and *Aedes aegypti* mosquitoes for flaviviruses). A closer examination showed that coronaviruses collectively favor 35 overused codons and 47 underused codons, while flaviviruses exhibit a more moderate bias, with 13 overused codons and 30 underused codons in Flavivirus, suggesting, again, stronger codon usage preferences in Coronavirus. Even if the three coronaviruses share similar codon usage patterns, SARS-CoV-2 demonstrates the most pronounced difference in codon usage compared to its human host. To a lower extent, ZIKV shows slightly more differences to its hosts than the two other flaviviruses.

To investigate the relationship between viral codon usage and human protein expression, we performed a hierarchical clustering and principal component analysis (PCA) based on RSCU values, incorporating human genes with varying protein abundance (as defined in Figure 1A) (Figure 2A,B). This analysis revealed distinct clustering patterns, effectively separating highly expressed human genes from the viral sequences. Notably, Coronavirus sequences clustered closely with poorly expressed human proteins, consistent with the PCA results shown in Figure 2B. In contrast, Flavivirus sequences occupied an intermediate position, indicating a codon usage pattern distinct from both highly expressed human proteins and from coronaviruses. Specifically, both SARS-CoV-2 and, to a lesser extent, ZIKV, demonstrate a prominent preference for codons read by tRNAs modified at the U34 wobble position, including ncm^5^U (noted as “o”), mcm^5^U (“oo”), or mcm^5^s^2^U (“ooo”). It is clearly the case for Ser^UCA^, Thr^ACA^, Pro^CCA^, Arg^AGA^, and Glu^CAA^ and, to a lesser degree, Lys^AAA^ and Gln^CAA^. While ZIKV displays approximately half the number of U34-sensitive codons as SARS-CoV-2, the trend remains significant. Given the crucial role of U34 tRNA modifications in efficient decoding, both viral genomes likely employ a shared mechanism to optimize translation by leveraging specific, and potentially limiting, host tRNA species.

### 2.2. tRNA Modification Remodeling During ZIKV and SARS-CoV-2 Infection

We previously observed distinct codon usage biases in ZIKV and SARS-CoV-2, raising the hypothesis that these viruses manipulate host tRNA modification patterns to optimize translation. To investigate this, we used mass spectrometry to measure tRNA modification levels in ZIKV-infected astrocytes and SARS-CoV-2-infected Caco2 cells at 24 h post-infection (Figure 3). Both viruses induced significant alterations in the host tRNA modification profile, notably affecting modifications at the wobble U34 position, although the specific direction and magnitude of these changes, as well as changes in other modifications (Figure 3A,C,D), differed significantly between ZIKV and SARS-CoV-2. This suggests that each virus has evolved distinct strategies to manipulate the U34 modification pathway (Figure 3B). These findings highlight the importance of U34 modifications during viral infection and suggest a key mechanism by which viruses fine-tune host translation.

### 2.3. Impaired U34 tRNA Generation Decreases Translation and Limits SARS-CoV-2 and ZIKV Replication

Elongator complex deficiency in Familial Dysautonomia impairs SARS-CoV-2 and ZIKV replication—While these data provide strong evidence for virus-induced changes in U34 tRNA modification, a direct assessment of the causal link between these modifications and viral translation remained to be established. To definitively assess the role of U34 modifications in viral replication, we next infected cells with impaired U34 modification pathways. We first made use of human primary fibroblasts derived from patients affected by Familial Dysautonomia (FD) (Figure 4A), a condition characterized by the loss of Elp1 expression (IKBKAP^-/-^) [34], a key component of the Elongator complex essential for the biosynthesis of mcm^5^s^2^U-modified tRNA (Figure 3B) [35]. We first validated the relevance of this cellular model by confirming the substantial decrease of ncm^5^U, mcm^5^U, and mcm^5^s^2^U tRNA levels in FD cells (Figure 4B), consistent with the intrinsic splicing defect in Elp1 mRNA previously reported in this disorder [36]. Upon infection with SARS-CoV-2, FD cells exhibited a marked reduction in viral infection levels compared to wild-type (*wt*) controls across all tested MOIs, with a substantial decrease of approximately 57–64% at higher MOIs (0.1 and 0.2) (Figure 4C). A similar trend was observed for ZIKV, with infection levels decreasing by 65–72% at high MOI (Figure 4D), further supporting the dependence of both viruses on an intact U34 tRNA modification pathway for efficient replication. Note that both FD and *wt* cells are naturally permissive to ZIKV, while they were previously transduced with an ACE2-expressing lentivector to ensure efficient SARS-CoV-2 viral entry, as fibroblasts are not naturally permissive to SARS-CoV-2 (Figure 4A).

Downstream U34 modification enzymes, ALKBH8 and CTU1, are essential for both SARS-CoV-2 and ZIKV replication—According to the U34 modification pathway (Figure 3B), alteration of the two enzymatic steps downstream of the Elongator complex (ELP1-6) step to generate mcm^5^s^2^U would also affect viral infection. We used an shRNA-based approach to examine the role of ALKBH8 and CTU1, key enzymes acting downstream of Elongator in the U34 pathway, in SARS-CoV-2 and ZIKV infection (Figure 5A). We confirmed the efficient knockdown of both ALKBH8 and CTU1 at the protein level (Figure 5B). ShRNAs targeting ALKBH8 and CTU1 both led to a significant reduction in SARS-CoV-2 and ZIKV infection levels compared to cells treated with the control shRNA (IRV1) (Figure 5C,D). Specifically, ALKBH8 and CTU1 knockdowns both resulted in an approximately 60% reduction in SARS-CoV-2 infection and a 90–95% reduction in ZIKV infection. These findings, consistent across six independent experiments, robustly demonstrate that SARS-CoV-2 and ZIKV infection relies on both ALKBH8 and CTU1/2 complex functions, in addition to the Elongator complex, to promote efficient U34 tRNA modifications and ensure optimal infection.

ELP1 knockout reduces both ZIKV infection rate and viral translation efficiency—To strengthen the tight relationship between U34 tRNA modifications and viral translation, we used a recombinant ZIKV expressing mCherry as a quantitative reporter of viral protein synthesis. Due to the lack of a comparable SARS-CoV-2 system with a fluorescent translation reporter, we could not study both viruses in parallel. To achieve higher infection levels than in primary fibroblasts (*wt* or FD), we used CRISPR/Cas9 to generate an ELP1 knockout (ELP1-KO) HeLa cell clone (Figure 6A), which showed a near-complete loss of ELP1 protein compared to control HeLa cells (Figure 6B). ZIKV infection levels, measured by the percentage of mCherry-positive cells, were significantly reduced in ELP1-KO cells compared to control cells at all MOIs tested (0.1, 2, and 5) (Figure 6C), ranging from approximately 42% to 47%. Consistent with the infection data, ZIKV viral translation, deduced from the mean mCherry intensity *per* infected cell, was also significantly reduced in ELP1-KO cells, with a decrease of approximately 39% to 47% across the same MOIs (Figure 6D). These results strongly suggest that ELP1, and consequently proper U34 tRNA modification, is essential for efficient ZIKV infection and viral translation. The substantial reduction in both infection levels and mCherry intensity underscores the importance of ELP1 for optimal ZIKV replication.

Collectively, these results suggest that the U34 tRNA modification pathway (comprising Elongator/ELP1, ALKBH8, and CTU1/2) significantly contributes to ZIKV and SARS-CoV-2 infection. They support the notion that these viruses manipulate the host tRNA epitranscriptome to adapt the host tRNA pool to their codon-biased viral genome through U34 modification to enhance their own translation.

## 3. Discussion

Our findings reveal a sophisticated strategy employed by SARS-CoV-2 and ZIKV to optimize their protein synthesis through the manipulation of host tRNA modifications, rather than evolving codon optimization towards their host codon usage. This conclusion is supported by both the computational analysis of viral codon usage and experimental evidence of tRNA modification dynamics during infection.

The codon usage analysis revealed that both viruses, particularly SARS-CoV-2, maintain a distinct codon bias from their human host. With CAI values lower for SARS-CoV-2 than for ZIKV, both viruses align more closely with poorly expressed human genes than with highly expressed ones. This apparent “non-optimization” of codon usage is particularly striking for SARS-CoV-2, which shows the most pronounced deviation from human codon preferences among the coronaviruses analyzed. The hierarchical clustering and PCA further confirmed this distinction, with viral sequences clustering separately from highly expressed human proteins. Notably, both viruses show a marked preference for codons requiring U34-modified tRNAs for efficient decoding, including Ser^UCA^, Thr^ACA^, Pro^CCA^, Arg^AGA^, and Glu^CAA^ codons. This preference is especially pronounced in SARS-CoV-2, which shows approximately twice as many instances of U34-sensitive codon usage compared to ZIKV. Rather than being a limitation, we proposed that this distinct codon usage represents an evolutionary strategy to manipulate host cell translation machinery. These findings align with and extend previous work demonstrating both the suboptimal translational adaptation of SARS-CoV-2 in infected human cells [25], and the preferential use of AA-ending codons in ZIKV that require U34-modified tRNAs for efficient decoding [37]. Building upon this previous bioinformatics analysis, this study significantly expands that work, offering a comparative analysis that extends to ZIKV and other flaviviruses, bringing broader insights into host–virus interactions across distinct RNA virus families.

Our experimental data strongly support the hypothesis that SARS-CoV-2 and ZIKV actively remodel the host tRNA epitranscriptome to accommodate their codon usage. We observed a significant increase in U34 tRNA modifications (ncm^5^U, mcm^5^U, and mcm^5^s^2^U) upon infection with both viruses. This manipulation of the host tRNA epitranscriptome, consistent with our previous findings on SARS-CoV-2 [25] and ZIKV [37], has also been reported in other RNA viruses with limited experimental backing, such as Chikungunya virus (CHIKV) [38] and Dengue virus (DENV) [39], suggesting a common viral strategy [40].

The importance of U34 tRNA modifications for efficient viral replication was further underscored by experiments using cells with impaired U34 tRNA generation. Fibroblasts from Familial Dysautonomia (FD) patients, lacking functional ELP1 and consequently exhibiting reduced mcm^5^s^2^U levels, showed a substantial decrease in both SARS-CoV-2 and ZIKV infection levels demonstrating a direct link between U34 tRNA modification status and viral replication capacity. This experimental design, utilizing human primary fibroblasts from FD patients along with a sex and age-matched wild-type control, allowed us to specifically investigate the role of U34 tRNA modifications in viral replication. Given that FD is primarily characterized by a splicing defect leading to a loss of Elp1 expression—a critical component of the Elongator complex essential for U34 tRNA modifications—the key difference between these cell lines lies in the expression of the ELP1 gene. This focused genetic distinction enabled us to effectively analyze the precise impact of compromised U34 modifications on viral processes, providing a well-controlled system for our studies.

To validate these findings and identify specific components of the modification pathway that are essential for viral infection, we performed shRNA targeted knockdowns of ALKBH8 and CTU1. The significant reduction in SARS-CoV-2 and ZIKV infection levels following these knockdowns corroborates our findings from FD cells and suggests that multiple components of the U34 modification pathway are important for viral infection. Although ALKBH8 or CTU1 knockdown cells may retain normal levels of ncm^5^U- or mcm^5^U-modified tRNAs, respectively, our data show that the presence of these intermediate modifications is not sufficient to support proper viral translation, highlighting the importance of the ultimate U34 modification, mcm^5^s^2^U. Knocking down essential tRNA biogenesis enzymes like ALKBH8 and CTU1 can affect global cellular translation and homeostasis and remains inherent to shRNA approaches. While our study focused on the direct impact of impaired U34 modifications on viral protein synthesis and replication, leading to significant reductions in viral infection, these broader cellular effects cannot be fully ruled out without a dedicated wide-spectrum study. Nevertheless, our findings clearly demonstrate the critical role of these specific tRNA modifications for efficient viral propagation.

To strengthen the link between U34 tRNA modifications and viral translation, we evaluated the ZIKV-directed expression of an mCherry reporter as an indicator of viral protein translation. Using ELP1-KO cells, we confirmed that ELP1, and by extension, proper U34 tRNA modification, is essential for efficient ZIKV infection and viral protein synthesis. The parallel reduction in both infection and translation strongly suggests that the availability of properly modified tRNAs directly impacts the ability of the virus to efficiently translate its genome and produce progeny virions.

In conclusion, through multiple complementary approaches, including U34 modification, deficient FD cells, targeted shRNA-mediated knockdowns, and ELP1 knockout, we demonstrate that disruption of the U34 modification pathway significantly impairs the replication efficiency of both SARS-CoV2 and ZIKV. We also establish the concomitant ability of both viruses to manipulate the host tRNA epitranscriptome, specifically targeting the U34 modification pathway. Given their pronounced codon usage bias relative to their human host, this strategy likely represents a novel viral adaptation to overcome translational bottlenecks. By actively enhancing the levels of specific tRNA modifications (ncm^5^U, mcm^5^U, and mcm^5^s^2^U), both viruses effectively fine-tune the host translation machinery to prioritize viral protein synthesis. This mechanism likely plays a crucial role in ensuring efficient viral replication and may represent a critical vulnerability that could be targeted for antiviral therapy. While our current study clearly demonstrated virus-induced alterations in U34 tRNA modification profiles through mass spectrometry, the precise quantification of the expression levels of the enzymes involved in the U34 tRNA modification pathway (e.g., Elongator complex, ALKBH8, and CTU1/2) could potentially reveal underlying mechanisms driving tRNA modification reprogramming. Exploring these upstream regulatory mechanisms and quantifying the viral infection-induced changes in the expression of these enzymes represents a crucial and valuable direction for future research, which will further deepen our understanding of this host–virus interplay. Future research should focus on elucidating the precise molecular mechanisms by which these viruses manipulate the U34 pathway and exploring whether pharmacological targeting of this pathway can effectively suppress viral replication in vivo. Furthermore, investigating the long-term consequences of viral-driven remodeling of the tRNA epitranscriptome on host cell homeostasis warrants further attention. Understanding the intricate interplay between viruses and the host translation machinery will be key to developing innovative and effective antiviral strategies.

## 4. Material and Methods

### 4.1. Bio-Informatics—Codon Analysis

Codon Adaptation Index (CAI) and Relative Synonymous Codon Usage (RSCU) calculations were carried out using CAIcal web-available tools (http://genomes.urv.es/CAIcal/ (accessed on 12 December 2024) [41]. Nc data were obtained using CAIcal and plotted with Rstudio-ggplot2 (version 2024.01). Principal component analysis was performed with ClustVis 2.0 (https://biit.cs.ut.ee/clustvis/, accessed on 8 November 2024) [42]. Codon frequencies were calculated using the Codon Utilization Tool (CUT) of HIVE-CUTs databases (https://dnahive.fda.gov/, accessed on 8 November 2024) [43]. Cluster analysis was carried out using Genesis 1.8.1 [44] and Cluster 3.0 [45], and visualizations were made with Java Treeview (Version: 2.11.4.0; https://www.jalview.org, accessed on 8 November 2024) [46]. Protein abundance levels were derived from the PaxDb database version 4.1 (https://pax-db.org, accessed on 8 November 2024) (accession numbers are available in Appendix A) [47]. Accession numbers of the virus sequences used in this report are listed in Appendix A and were downloaded from the NCBI database with the exception of SARS-CoV-2 FRA, which was downloaded from the European Virus Archive (http://www.european-virus-archive.com, accessed on 8 June 2020) and corresponded to the exact 2020 isolate from Paris-Ile-de-France that we used in our infection experiments.

### 4.2. Cells and Viruses

HeLa, A549, HEK293T, VeroE6, and Caco2 cells and human primary astrocytes (obtained from ScienCell (Carlsbad, CA, USA) were cultured in a DMEM Glutamax medium (GIBCO (Thermo Fisher Scientific, Waltham, MA, USA)), supplemented with Penicillin/Streptomycin and 5% of fetal calf serum. Patient primary fibroblasts were from the Coriell Institute (New Jersey, USA): GMO1652 derived from non-FD control (skin fibroblast (arm) from 11-year-old Caucasian female); GMO4959 derived from FD patient (skin fibroblast (arm) from 10-year-old Caucasian female). Fibroblasts were cultured as above. The SARS-CoV-2 was a French Ile-de-France isolate (www.european-virus-archive.com/virus/sars-cov-2-isolate-betacovfranceidf03722020 (accessed on 12 March 2024). Viral stocks were generated by amplification on VeroE6 cells (epithelial kidney of an African green monkey, ATCC CRL-1586). The supernatant was collected, filtered through a 0.45 µm membrane, and titrated using a TCID50 assay. Caco2 cells (epithelial colon adenocarcinoma, ATCC HTB-37) were used for tRNA modification quantification upon SARS-CoV-2 infection. For SARS-CoV-2 infection of primary fibroblasts or A549 cells, the cells were previously transduced with a lentiviral vector expressing ACE2 using the lentiviral construct RRL.sin.cPPT.SFFV/Ace2.WPRE (MT136) kindly provided by Caroline Goujon (Addgene plasmid # 145842). Seventy-two hours after transduction, accurate ACE2 expression was controlled by Western blots using anti-ACE2 antibody (Human ACE-2 Antibody, AF933, R&D systems (a Bio-Techne brand (Minneapolis, MN, USA)). ZIKV virus used was either the isolate from French Polynesia (PF13), generated from molecular clone from Mathew Evans [48], or the BeH8 (mCherry reporter) ZIKV replicon [49] from Andres Merits (Faculty of Science and Technology, University of Tartu, Estonia). Virus production and titration were performed in VeroE6 cells.

### 4.3. Quantification of tRNA Modifications by Mass Spectrometry (LC-MS/MS)

RNA preparations enriched in tRNAs were obtained using mirVana™ miRNA Isolation Kit (Thermo Fisher Scientific, Waltham, MA, USA). RNA samples were then digested by Nuclease P1 and treated by Alkaline phosphatase. Samples were then injected into an LC-MS/MS spectrometer. The nucleosides were separated by reverse phase ultra-performance liquid chromatography on a C18 column with online mass spectrometry detection using an Agilent 6490 triple-quadrupole LC mass spectrometer in multiple reactions monitoring (MRM) positive electrospray ionization (ESI) mode (Agilent Technologies, Santa Clara, CA, USA). Quantification was performed by comparison with the standard curve obtained from pure nucleoside standards running with the same batch of samples.

### 4.4. Gene Invalidation (CRISPR/Cas9 and shRNA)

The plasmids for CRISPR-Cas9 were obtained from the Montpellier Genomic Collection Platform (Biocampus, Montpellier, France). Guide RNA targeting ELP1 was designed using three online gRNA-optimizing software: CRISPR design (http://crispr.mit.edu.insb.bib.cnrs.fr (accessed on 8 September 2023)), CRISPR RNA Configurator (https://portals.broadinstitute.org/gppx/crispick/public (accessed on 16 June 2024),), and CRISPR gRNA Design tool (https://horizondiscovery.com/en/ordering-and-calculation-tools/crispr-design-tool, accessed on 16 June 2024). The guide RNA used was: 5′-GACTGTTGGAAACTATCACTGG-3′ (the PAM is underlined). The guide was cloned into pUC57 attB U6 gRNA vectors [50]. The generated plasmid pUC57 attB U6 gRNA was transfected into HeLa cells with Lipofectamine 2000, along with the pSpCas9(BB)-2A-GFP (PX458) plasmid [51]. At 6 h after transfection, the cells were trypsin treated and resuspended in a complete DMEM medium at 2 × 10^4^ cells per ml. Portions (200 μL) of the cell suspension (4 × 10^3^ cells) were used to inoculate 96-well plates and to isolate single cell-derived clones by serial dilution. Isolated green fluorescent protein (GFP)-positive clones were amplified and analyzed by Western blotting to check target gene expression. ShRNAs were Mission shRNAs from SIGMA cloned into plKO.1-puro with the following sequences: ALKBH8 (NCBI ID 91801, exon 12) 5′-CCGG**CAGGTGGGAAGGCACTCATTT**CTCGAG**AAATGAGTGCCTTCCCACCTG**TTTTTG-3′; CTU1 (NCBI ID 90353, exon 3) 5′-CCGG**CTTCTCCGAGGAGTGCGTCTA**CTCGAG**TAGACGCACTCCTCGGAGAAG**TTTTTG-3′, with bold sequences reverse complementary. Irrelevant shRNA sequence (IRV1): 5′-CCGG**GCGCGATAGCGCTAATAATTT**CTCGAG**AAATTATTAGCGCTATCGCGC**TTTTT-3′. Corresponding VSV-G pseudo-typed virus-like particles were produced in HEK293T cells as previously described [52] and used to transduce A549 cells. Individual clones were isolated after puromycin selection and analyzed by Western blotting to check target gene expression.

### 4.5. Assays for Viral Replication

Cells were lysed with the Luna cell-ready lysis module (New England Biolabs, Ipswich, MA, USA). The amplification reaction was run on a LightcyclerR 480 thermocycler (Roche Diagnostics (Indianapolis, IN, USA)) using the Luna Universal One-Step RT-qPCR kit (New England Biolabs) and SARS_For: 5′-ACAGGTACGTTAATAGTTAATAGCGT; SARS_Rev: 5′-ATATTGCAGCAGTACGCACACA; ZIKV_For: 5′-AGGATCATAGGTGATGAAGAAAAGT (hybridizes at the end of NS5 sequence); ZIKV_Rev: 5′-CCTGACAACATTAAGATTGGTGC (hybridizes in the 3′UTR region); GAPDH_For: 5′-GCTCACCGGCATGGCCTTTCGCGT; and GAPDH_Rev: 5′-TGGAGGAGTGGGTGTCGCTGTTGA primers. Each qPCR was performed in triplicate, and the means and standard deviations were calculated. Relative quantification of data obtained from RT-qPCR was used to determine changes in viral gene expression across multiple samples after normalization to the internal reference GAPDH gene. In cells infected with the ZIKV-mCherry translation reporter virus, infection level (% of mCherry-positive cells) and translation efficiency (relative mean of mCherry fluorescence intensity) were quantitated by flow cytometry on Novocyte (MRI CNRS facility, Montpellier, France).

## Figures and Tables

**Figure 1 ijms-26-07479-f001:**
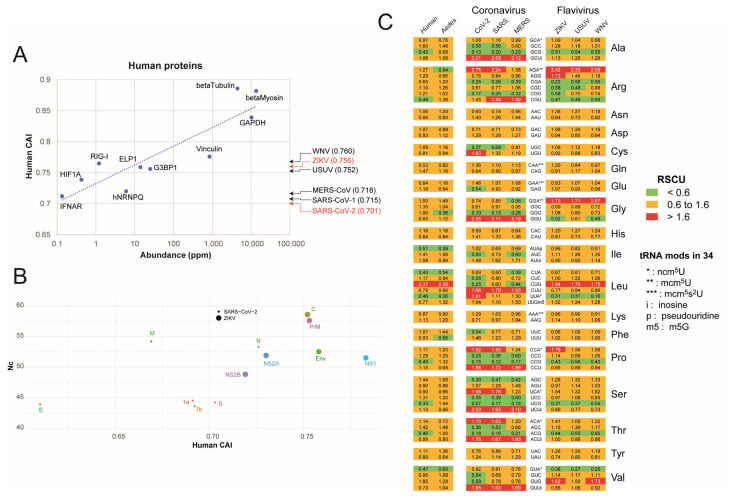
Codon Adaptation Index (CAI) and Relative Synonymous Codon Usage (RSCU) analysis of human genes and viral genomes**.** (**A**) Correlation between CAI and protein abundance (parts per million, ppm) for selected human genes. The dashed line represents a trendline visualizing the general relationship between these two variables. Highlighted human proteins serve as reference points for comparison with viral CAI values. Arrows indicate the CAI values for the analyzed viruses (SARS-CoV-2, SARS-CoV-1, MERS-CoV, ZIKV, USUV, and WNV). Note the logarithmic scale for abundance. The viral CAI values are shown in parentheses after the virus name. (**B**) Scatter plot of total number of codons (Nc) values versus human CAI for individual viral proteins of SARS- CoV-2 and ZIKV. (**C**) Heatmap showing the Relative Synonymous Codon Usage (RSCU) for select codons in the analyzed virus coding genomes and host genomes (human and *Aedes aegypti* mosquito). Red indicates RSCU > 1.6 (over-utilization), yellow indicates RSCU between 0.6 and 1.6, and green indicates RSCU < 0.6 (under-utilization). The modifications at the position U34 of corresponding tRNA anticodons are also indicated (* ncm^5^U, ** mcm^5^U, and *** mcm^5^s^2^U). “i”, “p”, and “m5” denote inosine, pseudouridine, and m^5^G modifications, respectively. The amino acids corresponding to each codon are listed on the right.

**Figure 2 ijms-26-07479-f002:**
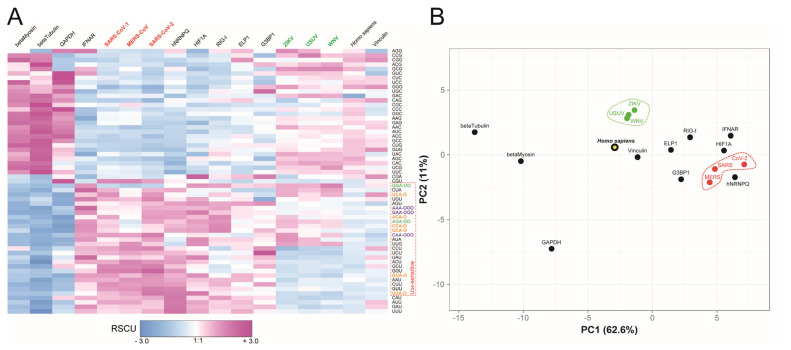
Codon usage bias in human genes and viral genomes. (**A**) Heatmap of Relative Synonymous Codon Usage (RSCU) values for selected codons across human genes, viral genomes, and *Homo sapiens* whole coding sequences (columns). Specific modifications at the U34 position of tRNA anticodons are indicated with “o” (ncm^5^U), “oo” (mcm^5^U), and “ooo” (mcm^5^s^2^U) to the right of the corresponding codons. (**B**) Principal component analysis (PCA) of RSCU values. The plot shows the distribution of human proteins (black circles), viral sequences (colored circles) and *Homo sapiens* genome (yellow/black circle), along the first two principal components (PC1 and PC2). The percentage of variance explained by each component is indicated in parentheses on the axes. Dashed circles highlight groupings of viral families.

**Figure 3 ijms-26-07479-f003:**
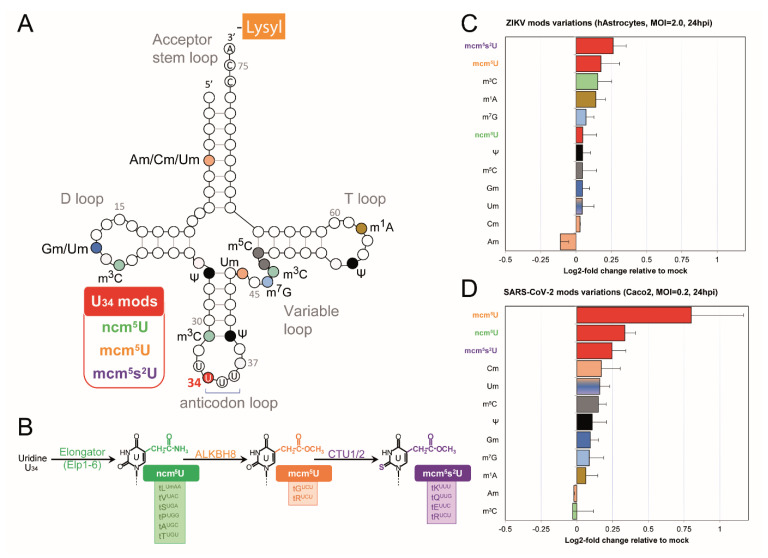
Changes in tRNA U34 modifications during ZIKV and SARS-CoV-2 infection. (**A**) Schematic representation of a typical tRNA molecule, illustrating the conserved secondary structure (cloverleaf) with acceptor stem (amino acid attachment) and anticodon loop (mRNA codon recognition). Variable, D, and T loops are also indicated. The positions of different modifications are color-coded across the tRNA, while the three modifications (mods) at the U34 site are specifically shown in red. (**B**) The enzymatic pathway of chemical modifications occurring at the wobble U34 position (ncm^5^U, mcm^5^U, and mcm^5^s^2^U) involves, successively, the Elongator complex (ELP1-6), ALKBH8, and CTU1/2 complex [33]. (**C**) Changes in tRNA modification levels detected in human astrocytes 24 h after ZIKV infection (MOI = 2). (**D**) Changes in tRNA modification levels in SARS-CoV-2 infected Caco2 cells at 24 h post-infection (MOI = 0.2). The (**C**,**D**) graphs show the log2-fold change in modification levels compared to mock-infected controls. Error bars indicate ± SD variations. Histogram of each modification is color-coded according to its location in panel (**A**). Um appears with a color gradient since it can be found at different positions (4, 18, 34, and 44).

**Figure 4 ijms-26-07479-f004:**
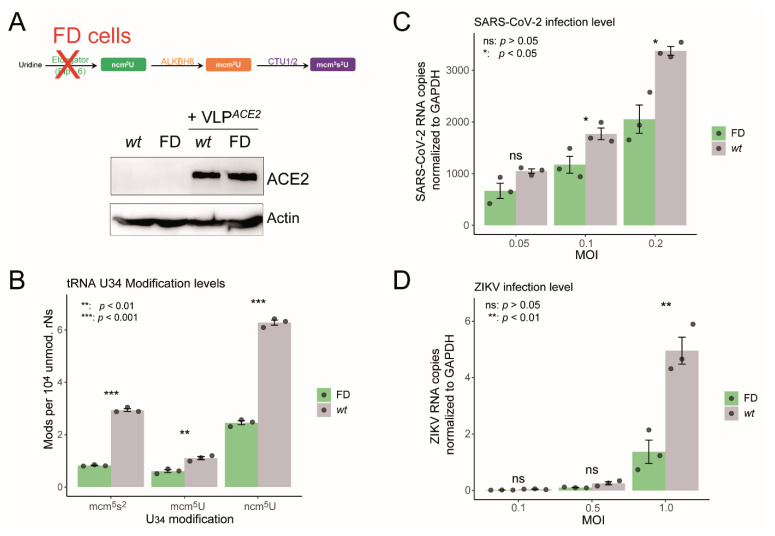
Elongator complex loss reduces SARS-CoV-2 and ZIKV infection. (**A**) Schematic representation of U34 tRNA modification pathway in FD (ELP1-deficient) cells. The Western blot shows ACE2 receptor and actin levels in cells transduced by ACE2-expressing lentivector (VLP^ACE2^). (**B**) Quantification of U34 modification levels (ncm^5^U, mcm^5^U, and mcm^5^s^2^U) in wild-type (*wt*) versus FD cells determined by mass spectrometry on tRNA-enriched RNA fractionations. (**C**) SARS-CoV-2 infection levels measured by RT-qPCR in *wt* and FD cells at different multiplicities of infection (MOI) 24 h post-infection. (**D**) ZIKV infection levels measured by RT-qPCR in *wt* and FD cells at different MOIs 48 h post-infection.

**Figure 5 ijms-26-07479-f005:**
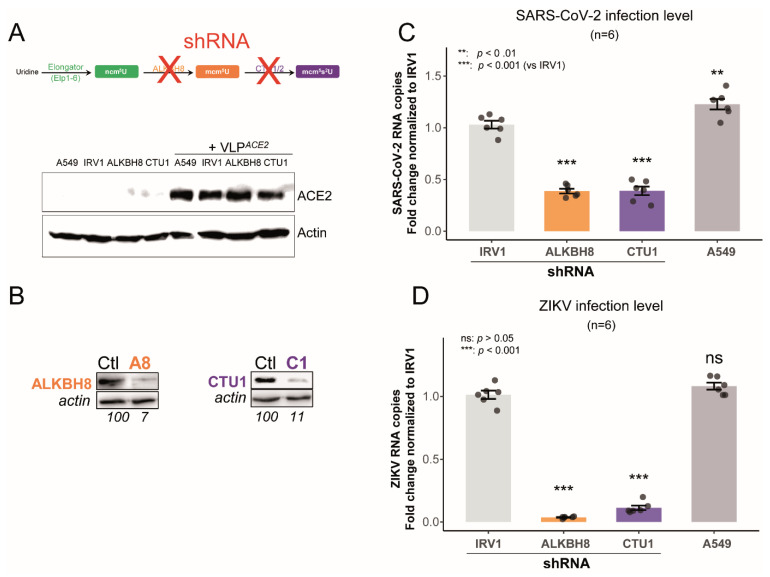
ALKBH8 and CTU1 knockdown individually impairs SARS-CoV-2 and ZIKV infection. (**A**) Schematic showing shRNA-mediated knockdown approach targeting U34 ALKBH8 and CTU1 components of U34 modification pathway. The Western blot analysis of ACE2 expression in A549 cells with and without lentiviral vector expressing ACE2 (VLP^ACE2^). (**B**) Western blot validation of ALKBH8 and CTU1 knockdown efficiency with corresponding actin loading controls. Numbers indicate relative protein levels. (**C**) SARS-CoV-2 infection levels measured by RT-qPCR following knockdown of the indicated genes 24 h post-infection. (**D**) ZIKV infection levels measured by RT-qPCR following knockdown of the indicated genes 48 h post-infection.

**Figure 6 ijms-26-07479-f006:**
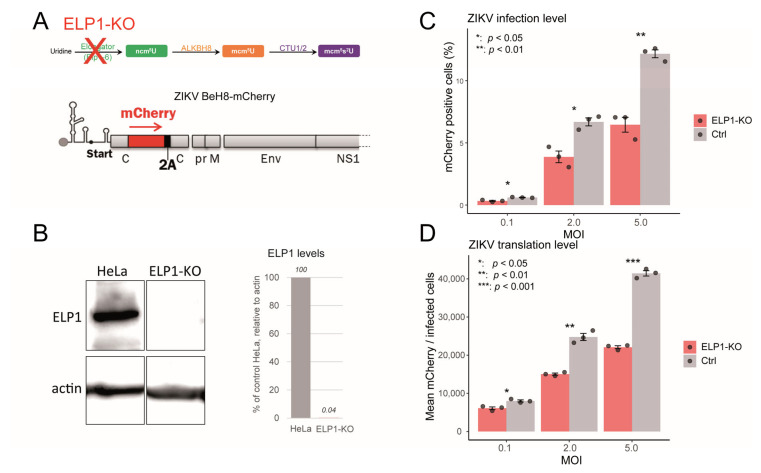
ELP1 knockout limits ZIKV infection and translation. (**A**) Schematic of U34 modification pathway in ELP1 knockout (Elp1-KO) cells. Schematic representation of ZIKV BeH8-mCherry 5′ genomic in which the mCherry coding region is followed by a “self-cleaving” 2A peptide and inserted between two tandem capsid (c) segments. (**B**) Western blot analysis comparing ELP1 protein levels between HeLa (control) and Elp1-KO cells with actin loading control. Graph shows quantification of relative ELP1 levels. (**C**) ZIKV infection rates measured by flow cytometry in control (HeLa) vs. Elp1-KO cells at different MOIs 24 h post-infection. (**D**) Quantification of ZIKV viral translation in control vs. ELP1-KO cells across different MOIs 24 h post-infection. Statistical significance and percent changes are indicated where applicable. Error bars represent standard error of the mean.

## Data Availability

All data generated or analyzed during this study are included in the manuscript or cited in reference.

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
