# Peer review of "tRNA Modifications: A Tale of Two Viruses—SARS-CoV-2 and ZIKV"

_ijms, 2025, doi:10.3390/ijms26157479_

Round 1

Reviewer 1 Report

Comments and Suggestions for Authors

An outstanding tale of Coronavirus (SARS-CoV2) and Zika virus (ZIKV) reveals similarity and differences of modification dynamics of U34 position of tRNA during infection of host cells by these two viruses focusing on multiple components of modification pathway. Based on well-designed experiments authors demonstrated that disruption of the U34 modification significantly impairs replication efficiency of the SARS-CoV2 and ZIKV. The revealed mechanisms clearly lead to the development of antiviral strategy in the nearest future.

In line 117 "calculated" instead of "caculated" should be written. 

Author Response

We appreciate the reviewers' time and insightful comments, which have significantly helped improve the clarity and rigor of our manuscript. Below, we address each point raised.

Reviewer 1 Comments

Comment: "In line 117 'calculated' instead of 'caculated' should be written."

Response: We thank the reviewer for pointing out this typographical error. We have corrected "caculated" to "calculated" in line 118.

Reviewer 2 Report

Comments and Suggestions for Authors

Current study by Eldin and Briant monitor tRNA modification dynamics in SARS-CoV-2 and ZIKV infected cells. The analysis show that SARS-CoV-2 and ZIKV infection enhance U34 tRNA modifications in the cells to possibly optimize viral translation. Further, the dependence of virus replication on U34 modification were analysed in cells impaired in U34 generation and decreased translation of viral proteins were observed. The manuscript is well written but I have some apprehensions about the experimental design:

1.      In section 2.3, what cells are being referred to as wild-type and would they only differ from the familial dystautonomia cells in a single variable?

2.      Similar to the concern raised in the comment 1, the knockdown of the essential enzyme in tRNA biogenesis pathway will have global affect in cellular translational landscape and homeostasis. Discuss any disadvantages of using this approach.

3.      Bioinformatics-based analyses discussed in this study have already been published in the context of SARS-CoV-2 viz. CAI and CUB analyses (10.3390/ijms252111614) by the same group.

4.      Did the authors quantify the viral infection-induced overexpression of enzymes involved in the U34 tRNA modification? Mechanistically, such a change in the levels of the enzyme would indicate virus-induced changes in host tRNA epitranscriptome machinery.

Minor comments:

Line 67: commas between Coronavirus and SARS-CoV-2, between Flavivirus and Zika Virus (ZIKV)

Line 101: advance> advancement

Line 396: tittered> tittered

Line 424-425: guide> guide RNA

Line 401-403 is not clear: “ACE expression is controlled on Western Blot”

Modification’s abbreviation ‘mods’ should be introduced in the text

Author Response

We appreciate the reviewers' time and insightful comments, which have significantly helped improve the clarity and rigor of our manuscript. Below, we address each point raised.

Reviewer 2 Comments

Comment 1: "In section 2.3, what cells are being referred to as wild-type and would they only differ from the familial dystautonomia cells in a single variable?"

Response: We thank the reviewer for this important clarification point. In Section 2.3, "wild-type" (wt) refers to human primary fibroblasts from an healthy donor that was used as control for the Familial Dysautonomia (FD) cells. As mentioned in the “Material and Methods” section, WT primary fibroblasts were derived from non-FD control (Skin fibroblast (arm) from 11-year-old Caucasian female) and FD primary fibroblasts were derived from FD patient (Skin fibroblast (arm) from 10-year-old Caucasian female).

As FD is characterized by a splicing defect leading to a loss of Elp1 expression, the primary difference between the FD cells and the wild-type cells is indeed the expression of the ELP1 gene, a key component of the Elongator complex essential for U34 tRNA modifications. This single variable difference allows us to specifically investigate the role of U34 modifications in viral replication. We will add this clarification to the manuscript.

Inserted Line 335 (->338): "This experimental design, utilizing human primary fibroblasts from Familial Dysautonomia (FD) patient and sex and an age-matched wild-type control, allowed us to specifically investigate the role of U34 tRNA modifications in viral replication. Given that FD is primarily characterized by a splicing defect leading to a loss of Elp1 expression—a critical component of the Elongator complex essential for U34 tRNA modifications—the key difference between these cells lies in the expression of the ELP1 gene. This focused genetic distinction enabled us to effectively analyze the precise impact of compromised U34 modifications on viral processes, providing a well-controlled system for our studies."

Comment 2: "Similar to the concern raised in the comment 1, the knockdown of the essential enzyme in tRNA biogenesis pathway will have global affect in cellular translational landscape and homeostasis. Discuss any disadvantages of using this approach."

Response: We agree with the reviewer that targeting essential enzymes in the tRNA biogenesis pathway, such as ALKBH8 and CTU1, can indeed have global effects on cellular translational landscape and homeostasis. We acknowledge this as a potential limitation of using shRNA-mediated knockdown of these enzymes. Our study primarily focuses on the direct impact of impaired U34 modifications on viral protein synthesis and replication, as evidenced by the significant reduction in viral infection levels observed upon knockdown of these enzymes. While we did not extensively characterize the global cellular effects, our findings strongly support the critical role of these specific modifications for viral propagation. We will add a discussion point in the manuscript to address the potential broader cellular impacts of these knockdowns, emphasizing that while global effects may occur, our results specifically demonstrate a viral dependence on this pathway for efficient translation and replication.

Inserted Line 354-361: “Knocking down essential tRNA biogenesis enzymes like ALKBH8 and CTU1 can affect global cellular translation and homeostasis and remains inherent to shRNA approaches. While our study focused on the direct impact of impaired U34 modifications on viral protein synthesis and replication, leading to significant reductions in viral infection, these broader cellular effects cannot be fully ruled out without a dedicated wide-spectrum study. Nevertheless, our findings clearly demonstrate the critical role of these specific tRNA modifications for efficient viral propagation.”

Comment 3: "Bioinformatics-based analyses discussed in this study have already been published in the context of SARS-CoV-2 viz. CAI and CUB analyses (10.3390/ijms252111614) by the same group."

Response: We thank the reviewer for this comment. While some bioinformatics analyses, such as CAI and CUB for SARS-CoV-2, have been previously published by our group [Eldin et al., 2024, IJMS, 10.3390/ijms252111614], this study notably expands the scope through a comparative analysis of SARS-CoV-2 with Zika virus (ZIKV) and other flaviviruses (West Nile virus and Usutu virus). This cross-family comparison offers significantly broader insights into host-virus interactions. Crucially, the core contribution of this manuscript lies in its novel experimental data, which definitively demonstrates the manipulation of the host tRNA epitranscriptome and its direct impact on viral replication for both SARS-CoV-2 and ZIKV. The bioinformatics analysis thus serves as an essential foundation and contextualization for these novel, experimentally validated findings within a broader comparative framework.

We will rephrase the relevant sections to ensure proper distinction and emphasis on the new findings.

Inserted Line 321-323: “Building upon our previous bioinformatics analysis (ref 25 and 37), this study significantly expands that work, offering a comparative analysis that extends to Zika Virus (ZIKV) and other flaviviruses, bringing broader insights into host-virus interactions across distinct RNA virus families.

Comment 4: "Did the authors quantify the viral infection-induced overexpression of enzymes involved in the U34 tRNA modification? Mechanistically, such a change in the levels of the enzyme would indicate virus-induced changes in host tRNA epitranscriptome machinery."

Response: We are grateful for the reviewer's comment regarding the quantification of U34 enzyme overexpression. Our study's primary focus was on quantifying changes in tRNA modification levels using mass spectrometry, which clearly revealed virus-induced alterations in the U34 tRNA modification profile. We acknowledge that we did not directly quantify the expression of enzymes within the U34 tRNA modification pathway (such as the Elongator complex, ALKBH8, or CTU1/2). This is a very pertinent mechanistic inquiry. We agree that such an analysis would significantly enhance our understanding of how the host tRNA epitranscriptome machinery is regulated during viral infection. As we are currently implementing specific approaches to uncover the molecular mechanisms behind the observed effects induced by both viruses, we will integrate this valuable suggestion into our discussion section as a key direction for future research, emphasizing the exploration of upstream regulatory mechanisms contributing to the observed tRNA modification changes.

Inserted Line 380-387: “While our current study clearly demonstrated virus-induced alterations in U34 tRNA modification profiles through mass spectrometry, the precise quantification of the expression levels of the enzymes involved in the U34 tRNA modification pathway (e.g., Elongator complex, ALKBH8, CTU1/2) could potentially reveal underlying mechanisms driving tRNA modification reprogramming. Exploring these upstream regulatory mechanisms and quantifying the viral infection-induced changes in the expression of these enzymes represents a crucial and valuable direction for future research, which will further deepen our understanding of this host-virus interplay.”

Minor Comments (Reviewer 2)

Comment: "Line 67: commas between Coronavirus and SARS-CoV-2, between Flavivirus and Zika Virus (ZIKV)"

Response: We have added the suggested commas in line 67 for clarity.

Comment: "Line 101: advance> advancement"

Response: We have corrected "advance" to "advancement" in line 101.

Comment: "Line 396: tittered>tittered"

Response: Line 424 We have replaced “tittered” by “titrated”.

Comment: "Line 424-425: guide> guide RNA"

Response: “guide” has been replaced by “guide RNA”

Comment: "Line 401-403 is not clear: "ACE expression is controlled on Western Blot""

Response: The reviewer might be referring to the Western blot analysis of ACE2 expression shown in Figure 4A and Figure 5A. The western blot analyses demonstrate the presence of ACE2 receptor levels in fibroblasts (Figure 4A) or in A549 cells (Figure 5A), previously transduced with an ACE2-expressing lentivector, ensuring efficient SARS-CoV-2 viral entry, as these cells are not naturally permissive to SARS-CoV-2. The Western blot confirms that ACE2 is expressed in these cells. We will rephrase the relevant text to make this point clearer. Line 430 now reads: “Seventy-two hours after transduction, accurate ACE2 expression was controlled by Western blots using anti-ACE2 antibody”.

Comment: "Modification's abbreviation 'mods' should be introduced in the text"

Response: We agree with the reviewer that the abbreviation "mods" should be formally introduced for consistency. The use of mods as an abbreviation of “modifications” has been specified in Figure 3’s legend.

Round 2

Reviewer 2 Report

Comments and Suggestions for Authors

All concerns have been answered in this version of the manuscript by the authors